# Rumen Microbial Predictors for Short-Chain Fatty Acid Levels and the Grass-Fed Regimen in Angus Cattle

**DOI:** 10.3390/ani12212995

**Published:** 2022-10-31

**Authors:** Jianan Liu, Ying Bai, Fang Liu, Richard A. Kohn, Daniel A. Tadesse, Saul Sarria, Robert W. Li, Jiuzhou Song

**Affiliations:** 1Department of Animal & Avian Sciences, University of Maryland, College Park, MD 20742, USA; 2College of Life Sciences and Food Engineering, Hebei University of Engineering, Handan 056038, China; 3College of Public Health, Zhengzhou University, Zhengzhou 450001, China; 4Food and Drug Administration, Center for Veterinary Medicine, Office of Research, Laurel, MD 20708, USA; 5United States Department of Agriculture, Agriculture Research Service, Animal Genomics and Improvement Laboratory, Beltsville, MD 20705, USA

**Keywords:** rumen, microbiome, grain-fed beef, grass-fed beef, Angus cattle, short-chain fatty acids

## Abstract

**Simple Summary:**

Grass-fed beef industry is booming in the USA. Compared to grain-fed, the rumen microbiome profiles and features in a grass-fed regimen have yet to be identified. In this study, we found that the rumen microbiome in the grass-fed cattle demonstrated greater species diversity and harbored significantly higher microbial alpha diversity than that of grain-fed cattle. The abundance of multiple unclassified genera, such as those belonging to Planctomycetes, LD1-PB3, SR1, *Lachnospira*, and *Sutterella*, were significantly enriched in the rumen of grass-fed steers. A rumen microbial predictor accurately distinguished the two feeding schemes. Multiple microbial signatures or balances strongly correlated with various levels of SCFA in the rumen. The results of this study provided deep insights into microbial interactions in the rumen under different feed schemes, which will help to develop rumen manipulation strategies to improve feed conversion ratios and average daily weight gains in beef practice.

**Abstract:**

The health benefits of grass-fed beef are well documented. However, the rumen microbiome features in beef steers raised in a grass-fed regimen have yet to be identified. This study examined the rumen microbiome profile in the feeding regimes. Our findings show that the rumen microbiome of the grass-fed cattle demonstrated greater species diversity and harbored significantly higher microbial alpha diversity, including multiple species richness and evenness indices, than the grain-fed cattle. Global network analysis unveiled that grass-fed cattle’s rumen microbial interaction networks had higher modularity, suggesting a more resilient and stable microbial community under this feeding regimen. Using the analysis of compositions of microbiomes with a bias correction (ANCOM-BC) algorithm, the abundance of multiple unclassified genera, such as those belonging to Planctomycetes, LD1-PB3, SR1, *Lachnospira*, and *Sutterella*, were significantly enriched in the rumen of grass-fed steers. *Sutterella* was also the critical genus able to distinguish the two feeding regimens by Random Forest. A rumen microbial predictor consisting of an unclassified genus in the candidate division SR1 (numerator) and an unclassified genus in the order Bacteroidales (denominator) accurately distinguished the two feeding schemes. Multiple microbial signatures or balances strongly correlated with various levels of SCFA in the rumen. For example, a balance represented by the log abundance ratio of *Sutterella* to *Desulfovibrio* was strongly associated with acetate-to-propionate proportions in the rumen (R^2^ = 0.87), which could be developed as a valuable biomarker for optimizing milk fat yield and cattle growth. Therefore, our findings provided novel insights into microbial interactions in the rumen under different feed schemes and their ecophysiological implications. These findings will help to develop rumen manipulation strategies to improve feed conversion ratios and average daily weight gains for grass- or pasture-fed cattle production.

## 1. Introduction

The rumen plays a critical role in host physiology and nutrition, as well as disease and production efficiency. In the rumen, microbial fermentation enables the conversion of plant fibers into small molecule products, particularly short-chain fatty acids, which get absorbed and digested by the animal [1]. This process makes it possible to utilize the solar energy stored in plant fibers by converting it into food products of the ruminant, such as milk and meat. Bacteria are the predominant microbes in the rumen, reaching a density of approximately 1010 to 1011 cells/mL of rumen liquid and occupying up to 90% of the total rumen microbial mass [2]. In addition, one crucial feature of ruminal bacteria fermentation is the production of short-chain fatty acids (SCFA) or volatile fatty acids (VFA) from plant fibers. SCFA contributes approximately 70% of the energy requirements of ruminants [2,3]. The principle SCFA in the rumen are acetate, propionate, and butyrate, which account for 95% of total SCFA produced, and are essential to rumen development and function [2,4]. The amount of these major SCFA in the rumen largely depends on diet [5]. It has been reported that diets that contain high fibrous components can produce a higher ratio of acetate-to-propionate, which generates more methane and, in turn, leads to wasted energy in cattle [2,5]. On the other hand, grain-fed diets tend to promote propionate and butyrate production due to increased numbers of relevant bacteria populations. Lactate is also an essential product of ruminal fermentation. The production systems relying on grass feeding and grain feeding affect the rumen’s SCFA and lactic acid levels [2,6]. Intermediate products are often related to ruminal acidosis in the high-grain feeding regimen, directly affecting animal health.

Recently, consumer demand for healthy beef products has been booming. Numerous studies in the past decades have demonstrated the health benefits of grass-fed beef, including improved fatty acid profiles and antioxidant contents [7]. Grass-fed beef contains a significantly higher total conjugated linoleic acid (CLA) and omega-3 polyunsaturated fatty acids. Further, increased consumer consciousness for animal welfare and environmentally friendly animal products has increased the demand. Consequently, the price premium for grass-fed beef has increased by approximately 70% over conventional grain-fed beef. As a result, grass-fed beef’s compound annual growth rate has been forecasted to increase by over 6% between 2020 to 2024, reaching a total global market size of $14.50 billion (https://www.businesswire.com (accessed on 10 March 2021)). Although we found the differences in beef quality and metabolites between those fed on the two diet styles, the similarities and differences in rumen microbial composition and species interaction between grass-fed and grain-fed cattle have not been systemically evaluated. Furthermore, the metabolite and microbial predictors or biomarkers associated with the grass-fed regimen have yet to be identified. In this study, we characterized the microbial interactions in the rumen using advanced algorithms and compared global microbial networks between grass-fed and grain-fed production systems. We also attempted to identify rumen metabolites, microbial features, and biomarkers with high predictive power for grass-fed cattle. Our findings will provide deep insight into enhancing production efficiency in the grass-feeding regimen.

## 2. Materials and Methods

### 2.1. Experimental Design and Sample Collection

Eighteen steers at the Wye Angus beef cattle herd of the University of Maryland were used for this study. Two types of feeding methods, grain-fed and free-range grass-fed, were used in this herd as the two experimental groups. The grain-fed group received a finishing diet consisting of silage corn, shelled corn, soybeans, and trace minerals. The grass-fed group had free access to grazed alfalfa and alfalfa, as well as bailage during the cold season. The alfalfa crop did not use fertilizers, pesticides, or other artificial chemicals. Grass-fed cattle in this herd were not fed any animal, agricultural or industrial by-products and received no grain. The chemical composition and digestible energy values of the feed used in the experiment were described [8]. The date of birth, birth weight, dam and sire information in spring were recorded. Every 24 to 28 days, body weight was measured to calculate average daily gain (ADG). After reaching the market weight of around 1000 lbs, the Angus steers, grass-fed animals at 22 months old and grain-fed animals at 16 months old, were slaughtered; and the liquid rumen samples were immediately collected and stored at −80 °C for microbial DNA extraction.

### 2.2. Measurements of Short-Chain Fatty Acids and Lactate

The levels of VFA and lactic acids in the rumen liquid were measured. A volume of 225 mL deionized water was added to 25 g of thawed digesta samples. The samples were mixed well by vortexing for 2 min. The samples were then centrifuged; and the clear supernatant was kept for gas chromatography to measure acetic, propionic, butyric, and iso-butyric acids. The aliquot of extracts was mixed at a 1:1 ratio with 0.06 M oxalic acid containing 100 ppm trimethylacetic acid as an internal standard. Samples were injected into a Perkin Elmer Autosystem XL Gas Chromatograph containing a Supelco packed column with the following specifications: 2 mm × 2 mm Tightspec ID, 4% Carbowax 20 M phase on 80/120 Carbopack B-DA. VFA concentrations were determined and expressed as parts per million (ppm). A Biochemistry Analyzer, YSI 2700 SELECT (YSI Incorporated Life Sciences, Yellow Springs, OH, USA), equipped with an L-Lactate membrane, was used for lactic acid measurements. Samples were injected into the sample chamber of the YSI Analyzer, where L-Lactate diffused into a membrane containing L-Lactate oxidase. The L-Lactate was immediately oxidized to hydrogen peroxide and pyruvate. The hydrogen peroxide was detected amperometrically at the platinum electrode surface. The current flow at the electrode is directly proportional to the hydrogen peroxide concentration, and hence to the L-Lactate concentration. Total lactic acid is determined by multiplying L-Lactate by 2.0.

### 2.3. DNA Extraction and 16S rRNA Gene Sequencing

Total DNA extraction and 16S rRNA gene sequencing was performed as described previously by Liu et al. [9]. Briefly, the QIAamp DNA stool kit (Qiagen, Valencia, CA, USA) was used with one protocol modification that replaced the lysis procedure with an eight-minute 95 °C incubation in a water bath. DNA concentration was then measured using Qubit and Agilent (Thermo Fisher Scientific, Waltham, MA, USA). Hypervariable V3-V4 regions of the 16S rRNA gene were amplified using PCR from 20 ng of total DNA with PAGE-purified Illumina platform-compatible adaptor oligos. The primer sequences were as follows: forward primer, 341/357F, CCTACGGGNGGCWGCAG; reverse primer, 805/785R: GACTACHVGGGTATCTAATCC. Amplicons were purified using Agencourt AMPure XP bead kits (Beckman Coulter Genomics, Danvers, MA, USA), and their concentration and size were determined using BioAnalyzer 2000 (Agilent, Palo Alto, CA, USA). The library pool was sequenced in an Illumina MiSeq sequencer with an Illumina MiSeq Reagent V3 Kit according to the manufacture’s protocol.

### 2.4. Bioinformatics and Statistical Analyses

Raw sequences were first analyzed with FastQC (version 0.11.5) to examine the quality of sequencing. The QIIME2 pipeline [10] was used to analyze the 16S rRNA gene sequence data. The DADA2 [11] software package was used to denoise and remove sequencing errors from Illumina paired-end amplicon sequencing reads. Both R1 and R2 reads were trimmed for 21 bp to remove primers and low-quality base pairs. DADA2 allows for correcting sequencing errors, removing chimeras, and deduplicating [11]. The operational taxonomic unit (OTU) was clustered at 100% similarity, and the resulted features were also known as Amplicon Sequence Variants (ASVs). Taxonomic assignment was performed based on the Greengenes database v13.8 [12]. Alpha diversity indices obtained from QIIME2 were further tested by Wilcoxon rank-sum test in R. The beta diversity was examined by PCoA using Bray–Curtis dissimilarity and tested for significance using PERMANOVA (Permutational Multivariate Analysis of Variance) in the vegan [13]. Differential analysis of taxa between the two groups was performed with ANCOM-BC [14]. To further obtain essential features of the two groups, Random Forest (RF) classification was performed with the group as the dependent variable using Random Forest package v4.6-14 [15]. The number of trees (ntree) in the forest was set to 801, and the number of features randomly sampled for each tree (mtry) split was 12. For the rumen microbial network analysis of grass-fed and grain-fed cattle, NetCoMi (Network Comparison for Microbiome data) was used [16]. A conditional dependence method, SPRING [17], was applied in the network construction step. Negative associations were handled with a “signed” method when sparsified associations were transformed into dissimilarities. The FDR of the network comparison step was controlled by an adaptive Benjamini–Hochberg method. Permutation was used with *n* = 1000. PICRUSt2 [13] was used with default parameters to predict KEGG gene families and functional categories between grass-fed and grain-fed groups based on 16S rRNA marker gene sequences. LEfSe was used to identify pathways that most likely explain the difference between two groups [18]. Welch’s *t* test was used for differential statistical analysis of SCFA between grass-fed and grain-fed groups. Correlations between the abundance at the genus level and SCFA were analyzed using the Spearman correlation with FDR adjustment for *p* values. 

## 3. Results

### 3.1. The Differences in Rumen Microbial Composition between Grass-Fed and Grain-Fed Cattle

In total, there were 8671 OTUs identified in the rumen using QIIME2. Of them, 5861 were identified in grass-fed cattle, while 4450 were found to be in the grain-fed group. However, only 1640 OTUs were shared between groups (Figure 1A). Bacteroidetes, Firmicutes, Proteobacteria, Fibrobacteres, and Tenericutes were the most abundant bacterial phyla. The stacked bar plot showed the top abundant phyla of the two groups; phyla with relative abundance less than 1% were grouped as “Others” (Figure 1B). These OTUs represent approximately 44 classes, 75 orders, 105 families, and 157 genera (Appendix A). 

Alpha diversity indices such as Ace (Abundance-based Coverage Estimator), Chao1, Shannon, and Simpson, were calculated from QIIME2 and analyzed by the Wilcoxon rank-sum test between the two groups to determine the significance of group comparisons (Figure 2B). The rumen in the grass-fed cattle showed significantly higher microbial diversity than its grain-fed counterparts (*p* < 0.05). Beta diversity was analyzed using Principal Component Analysis (PCoA) based on Bray–Curtis dissimilarity (top 20 genera). As Figure 2A shows, there was a clear separation between the two feed regimens. Permutational Multivariate Analysis of Variance (PERMANOVA) was used to test the significance. The effect of the feeding regimens alone explained 68.6% of the variation in beta diversity in rumen bacterial communities (permutation *p* = 0.001). 

### 3.2. Microbial Taxa Significantly Correlated with Rumen Short-Chain Fatty Acid Contents

Welch’s modified two population *t* test was used to examine the differences in VFA and lactate contents in the rumen between grass-fed and grain-fed cattle (Table 1). Compared to grain-fed cattle, the rumen in grass-fed cattle had a significantly lower acetate, propionate, butyrate, isobutyrate, and total SCFA (*p* < 0.05). However, while the grain-fed group had a numerically higher lactate concentration, it failed to reach the statistically significant level (*p* > 0.05). At the genus level, correlation analysis was performed to examine the correlation between the rumen microbiome and SCFA. We computed the Spearman correlation coefficient between the abundance of each genus with acetate, propionate, butyrate, and total level of SCFA, respectively. The results from 20 abundant genera were shown in the correlation heatmap (Figure 3). Notably, all SCFA displayed a significantly negative correlation with unclassified genera in the families *RFP12* and *BS11*, and a positive correlation with an unclassified genus in *Clostridiales* (*p* < 0.01). Intriguingly, several standard and core rumina microbiome components, such as *Prevotella*, *Butyrivibrio*, and *Fibrobacter*, were not correlated with any SCFA measurements, a phenomenon which needs to be studied further. At the same time, the abundance of *Ruminococcus* was associated only with propionate and total SCFA, but not with acetate and butyrate levels.

### 3.3. Differentially Abundant Taxa and Predicted Function Categories between the Two Feeding Regimens 

Significant taxa displaying differential abundance levels between the two feeding regimens were detected using the analysis of compositions of microbiomes with bias correction (ANCOM-BC) algorithm [14], which deals with the compositionality of microbiome data while correcting for the presence of bias. Of 64 genera showing significantly differential abundance levels between grass-fed and grain-fed groups, 44 were more abundant in the grass-fed group than the grain-fed group. At the same time, 20 were more productive in the latter (Appendix A). The 40 most significant genera were plotted in Figure 4. Log fold change is shown with standard deviation as an error bar for each taxon, and colors annotated the group. *Amoebophilus*, *Acetobacter*, *Megasphaera*, *Succinomonas*, and an unclassified genus in the family Succinivibrionaceae were among the most enriched in the grain-fed group. *Methanobrevibacter*, which contains strictly anaerobic archaea and produces methane, was also more abundant in grain-fed cattle. On the other hand, *Bulleidia*, *Lachnospira*, *Dehalobacterium*, and several unclassified genera in the families of Planctomycetes, whereas SR1 and LD1 were more abundant in the grass-fed group. Certain bacteria in the genus Planctomycetes can hydrolyze cellulose and hemicellulose [19]. Therefore, a significant increase in the abundance of this group may reflect the changes in the contents of structural carbohydrates enriched in the grass-fed. Further, among the genera with significantly higher abundance in the grass-fed cattle, unclassified taxa were notably more numerous, suggesting that more microbial diversity in the rumen of grass-fed cattle has yet to be captured. 

A Random Forest classification model was used to distinguish the two feeding regimens (mtry = 12, ntree = 801; Figure 5). A 100% classification accuracy between the two groups was achieved. Among the top 15 essential genera that contribute to the discrimination of the grass-fed from grain-fed groups, *Sutterella* and two unclassified genera assigned to the families M2PT2-76 and Mycoplasmataceae, respectively, were the three most features in distinguishing grass-fed from those grain-fed cattle. 

Further, the microbial functional categories or KEGG pathways were calculated using PICRUSt2 based on the ASV data derived from QIIME2 [13]. The pathways displaying a significant difference in the relative abundance were identified using the LEfSe algorithm [18]. Among the 310 KEGG pathways identified, 36 showed significant differences in the relative abundance between the two feeding regimens (Figure 6). As Figure 6 shows, the pathways related to carbon metabolism, fatty acid metabolism, fatty acid biosynthesis and degradation, CTA cycles, and lipopolysaccharide biosynthesis, were most enriched in the rumen of grass-fed cattle. On the other hand, ABC transporters, two-component systems, quorum sensing, and pathways related to biofilm formation were significantly higher in the grain-fed than grass-fed cattle. In addition, lysine metabolism displayed a divergence between the two feeding regimens: lysine degradation was enriched in the rumen of grass-fed cattle, whereas lysine biosynthesis was higher in grain-fed cattle.

### 3.4. Rumen Microbial Interaction Networks

Microbial interaction networks were constructed and compared between grass- and grain-fed cattle using a recently developed algorithm, NetCoMi [16]. Eigenvector centrality was used for determining hubs and scaling the node size. Networks were first constructed using a conditional dependence algorithm, SPRING, with the same layout in each instance to achieve a graphical comparison between the two groups. The “signed” distance metric transformed the calculated partial correlations into dissimilarities, and the edge weights were calculated from similarities between genera. Greedy modularity optimization was applied to determine clusters, which were annotated by different colors. Clusters were represented with the same color in two networks if they shared a minimum of two taxa. Positive associations were represented as green, and negative associations were shown as red. The top 10 genera with the highest absolute group differences in degree and eigenvector centrality are shown in Table 2. The term degree represented the number of adjacent nodes and was normalized to [0, 1]. Highly different eigenvector centralities between the two groups denoted genera with highly different node sizes in the network plots, even if the *p* value was not significant. For example, *Anaeroplasma* had a higher eigenvector centrality in the grain-fed group than the grass-fed group. This genus was also detected as a network hub in the grain-fed cattle (Figure 7). However, it did not act as a network hub in the grass-fed network.

Furthermore, we also used the Jaccard similarity coefficient to evaluate the network similarity. The statistic is used for gauging the similarity and diversity of sample sets. The Jaccard index value (j) of the networks is shown in Table 3, which described the similarity of the sets of most central nodes and hub taxa between the two networks. “The NetCoMi algorithm calculated most central” nodes as the features with a centrality value more significant than the empirical 75% quantile [16]. The index value 0 indicated that the two sets were completely different, while 1 meant that the sets were precisely equal. *p* (J ≤ j) is defined as the probability that Jaccard’s index takes a value less than or equal to the calculated index j for the total number of features in both sets, while *p* (J ≥ j) is the probability greater than or equal to j [16]. The results in Table 3 suggested that the sets of central nodes between the two networks derived from the grass and grain feeding regimens were significantly different.

To further quantify the network characteristics, global network attributes were tested for significance (Table 2). Network metrics and centrality measures of the networks for the two groups were compared using multiple testing with an adaptive Benjamini–Hochberg method with permutation. A notable global network feature was modularity. The global networks in the rumen of grass-fed cattle were highly modular compared to grain-fed cattle (*p* = 0.003). As an important indicator of ecosystem resilience, a modular structure of microbial interactions in the rumen of grass-fed cattle may represent the dynamical stability of rumen microbial communities. The microbiota in the rumen in different diet regimes focus on developing a specialized and relatively autonomous strategic niche. These discoveries provide new insights into the colonization of bacterial communities in the rumen that may be useful in designing strategies to promote the colonization of target communities to improve functional development and emission.

By comparing the network patterns, network hubs and types of associations between the two groups can be observed. As Figure 7 depicts, the microbial interaction patterns between the two feeding regimens were markedly different. Different hubs between the two networks were apparent. Unliked the grass-fed network, unclassified genera in Acholeplasmatales (Acholepla) and Endomicrobia (Endomicro) acted as network hubs in the grain-fed network. Further, there was an unclassified genus in BS11 in the network derived from grass-fed cattle which was strongly and positively associated with several unclassified genera in the families Prevotellaceae, Paraprevotellaceae and Candidatus. Two genera in the Paraprevotellaceae and Candidatus also acted as network hubs. As a group of common gut microbes able to ferment hemicellulosic monomers to acetate and butyrate in ruminants [20], BS11 was significantly enriched in the rumen of grass-fed cattle (Figure 3). Grass-fed diets contain increased amounts of hemicellulose and lignin.

### 3.5. Identification of Microbial Predictors or Balances Using the Selbal Algorithm

A rumen microbial signature with high accuracy in predicting the two feeding regimens was identified using the selbal algorithm as shown in Figure 8. This microbial predictor consisting of an unclassified genus in the candidate division SR1 (numerator) and an unclassified genus in the order Bacteroidales (denominator) accurately predicted grass- and grain-fed schemes with an area under the receiver operating characteristic curve (AUC) of 1.0 (mean cross-validation or CV-AUC = 0.935) (Figure 8A). This ruminal signature in grass-fed cattle tended to have a significantly higher balance value. The negative balance values indicated that the relative abundance of the denominator was much higher than that of the numerator. Several microbial signatures were also detected with predictive accuracy for total SCFA and its major components, particularly butyrate and propionate. The same microbial predictor consisting of *Clostridium* (numerator) and an unclassified genus in the uncultured bacterial group PHOS-HD29 (denominator) was positively correlated with ruminal propionate (Figure 8B) and butyrate levels (R^2^ = 0.783 and 0.640, respectively. Similarly, the log-ratio between *Pyramidobacter* (numerator) and an unclassified genus in the group PHOS-HD29 (denominator) was significantly correlated with total SCFA concentrations in the rumen (R^2^ = 0.592). The genus Pyramidobacter, a member of the phylum Synergistetes, includes a group of newly described anaerobic, non-motile, asaccharolytic bacilli that produce acetic and isovaleric acids and small amounts of propionic or isobutyrate, among others [21]. Interestingly, a rumen microbial signature consisting of Sutterella (numerator) and *Desulfovibrio* (denominator) was strongly correlated with the acetate-to-propionate ratio in the rumen (R^2^ = 0.868), regardless of the feeding schemes. Expectedly, a microbial signature represented by the log abundance ratios of *Desulfovibrio* and an unclassified genus in BS11 to *Sutterella* was strongly correlated with the inverse of the acetate-to-propionate ratios (i.e., the propionate-to-acetate ratio) (R^2^ = 0.919).

## 4. Discussion

Grass-fed cattle production represents a small but rapidly growing segment of the US cattle industry, primarily driven by increasing consumer demands for healthier beef products and consumer consciousness for animal welfare and the environment. A recent model suggests that, to maintain the same quantity of beef produced, a nationwide shift to exclusively grass-fed beef would require increasing the national cattle herd by 30% [22]. When this consumption trend is expected to continue, an increased environmental burden [23,24], including higher methane emissions [25], resulting from such a swift should be carefully considered. Grass-fed cattle tend to have a significantly lower average daily weight gain (ADG) and reduced age–weight index, as defined as a ratio of estimated carcass weight (lb) divided by age at slaughter (months), likely due to less efficient feed conversion ratios (FCR) for cattle grazing on pasture. In our study, significantly lower level of total SCFA and its major components, acetate, propionate, and butyrate, were observed in the rumen liquid of grass-fed cattle than grain-fed cattle. As a major energy source, SCFA provides greater than 70% of the metabolizable energy supply in ruminants. The subdued production of total SCFA in the rumen of grass-fed cattle may result in less efficient FCR and subsequently reduced ADG. In the follow-up experiment, we will investigate the possible correlation between ruminal SCFA, ADG, and carcass weight gain in several critical beef cattle production stages under grass and grain feeding regimens. Furthermore, increasing feed conversion via selective breeding or rumen manipulation may present a pragmatic solution to improve production efficiency in the grass feeding regimen. Understanding the essential ruminal microbial and fermentation characteristics of the grass-fed regimen represents this goal’s first step.

As an essential fermentation chamber, the rumen epithelium and compartmentation hold keys to the efficient microbial breakdown of dietary fibers. The rumen microbial community plays a critical role in the process. This study demonstrated that the rumen of grass-fed cattle harbored a significantly higher microbial alpha diversity than that of grain-fed cattle. Both bacterial species richness and evenness-related diversity indices were significantly higher in the rumen of grass-fed cattle. Moreover, a substantial portion of rumen microbial taxa in the grass-fed cattle remains unclassified, suggesting that there exists a dazzling array of previously uncharacterized microbial diversity under this feeding regimen. The reduction in alpha diversity in the rumen of grain-fed cattle was likely due to limited substrate availability to the bacteria that ferment structural carbohydrates, and subsequently a lower pH [26]. The beta diversity based on Bray–Curtis dissimilarity metrics displayed a clear separation between the two feed regimens, and the feeding scheme can explain up to 69% of the variation. The contemporary concept of biodiversity includes not just richness and evenness, but also phylogenetic diversity and special microbial interactions. Network analysis using a conditional dependence model detected significantly higher modularity in the rumen of grass-fed cattle, suggesting that grass feeding likely resulted in a more resilient and stable rumen microbial community than grain feeding. It is conceivable that such a stable microbial community will translate to a healthier gut environment conducive to improving animal welfare.

It is well known that a higher ratio of fermentable carbohydrates in cattle diets promotes SCFA production in the rumen, especially that of propionate and butyrate [27,28]. The acetate-to-propionate ratio tends to be higher in high-forage diets [28]. Our results were in agreement with the previous reports. We also observed that the concentration of propionate was relatively more significant than that of acetate. As a result, the rumen of grain-fed cattle had a lower acetate: propionate ratio. Moreover, acetate is mainly produced via microbial fermentation of fibrous components [27]. To examine the correlation between rumen microbial taxa and SCFA, we computed the Spearman correlation coefficient between the abundance of key genera with acetate, propionate, butyrate, and total SCFA levels. Notably, all major SCFA components showed a significantly negative correlation with two unclassified genera in the families RFP12 and BS11 and a positive correlation with an unclassified genus in Clostridiales (*p* < 0.05). The order Clostridiales harbors some key butyrate producers [29,30]. We speculate that characterization and subsequent development of these unknown rumen bacteria in Clostridiales as probiotics may enhance ruminal SCFA production, likely improving the ADG of grass-fed cattle.

Cattle consuming the grain diet had higher representation of the starch-fermenting bacteria Succinivibionaceae and Succinimonas, which produce succinate, acetate and lactate. Although protists were not quantified, the animals on grain-fed diets had more of the amoeba-utilizing Amoebophilus. Lactate-utilizing bacteria, including Megasphaera (strictly anaerobic) and Acetobacter (facultative aerobic), were also found at higher levels in the grain diets. The lactate concentration tended to be higher on the concentrate diet, but elevation of Megasphaera in particular suggests a potential shift through the lactate pathway on the high grain diet [31]. On the high grass diets, genera associated with fiber digestion were elevated. The highest increase compared with the grain-fed was for Planctomycetes. Switching cattle from a high-forage to a high-starch diet resulted in a decrease in the rumen pH value because of SCFA accumulation in the rumen [32,33]. The increase in nonstructural carbohydrates in cattle’s diet during gradual grain adaptation is frequently associated with a change in the relevant microbial population in the rumen. The family Succinivibrionaceae functions as a major succinate producer [34], which competes with methanogens for hydrogen required to make succinate, a precursor for propionate. The increase in the rumen Succinivibrionaceae populations may be associated with greater starch availability [35]. High-grain diets appear to favor the growth of these bacterial populations [34,36]. In this study, both an unclassified genus and *Succinimonas* in the family Succinivibrionaceae were significantly enriched in the rumen of grain-fed cattle (Figure 8) compared to grass-fed cattle, which directly or indirectly contributes to the increased production of propionate under this feeding scheme. Further, at the same time the family RFP12 remains poorly characterized. It is found to be a key member of gut microbes in horses and ruminants. This group is designated a core-heritable microbiota member in dairy cows corresponding to levels of methane, rumen, blood metabolites and milk production [37]. A previous report shows that steers fed sainfoin silage had lower concentrations of branched-chain VFA (*p* < 0.05), while the proportion of the phylum Verrucomicrobia was higher [38]. In our study, the family RFP12 contributes to >80% of the abundance of Verrucomicrobia. Our data provided a snapshot of complex interactions among diets, SCFA production, and critical microbial constituents. Grain-fed diets affect SCFA production, which in turn interacts with lactate-associating bacteria in the rumen. A high amount of SCFA production first reduced ruminal pH in the grain-fed cattle, promoting the expansion of both lactate-producing and lactate-utilizing microbial populations. On the other hand, the genus Mycoplasma is linked to a variety of inflammatory diseases in cattle. It has been reported that the alterations in the rumen environment and rumen epithelial-associated bacterial communities were induced by high-grain feeding, which may result in damage and local inflammation in the rumen epithelium [39]. With wheat and barley as the main concentrate ingredients and hay as the primary forage, the dietary forage-to-concentrate ratios of less than 39:61 appeared to decrease the rumen pH sufficiently to activate an inflammatory response [40]. Our observation that an unclassified genus in the family Mycoplasmataceae was significantly enriched in the rumen of grain-fed cattle suggests that this feeding regimen may have potential negative implications on the health and wellbeing of cattle. Indeed, this taxon was among the essential features identified using Random Forest to distinguish the two feeding regimens (Figure 5).

As a significant substrate for *de novo* FA synthesis, acetate plays an important role in ruminant physiology and nutrition. Acetate increases milk fat and milk fat concentration in a dose-dependent manner in lactating cows [41]. On the other hand, as a propionate is a gluconeogenic VFA, propionate can increase energy availability to the mammary gland [42]. Its fermentation pathway is distinct from those resulting in acetate and butyrate synthsis by not liberating hydrogen. As a result, increasing propionate production may reduce methane emissions from the rumen. Indeed, a positive correlation between methane production and the ratio of ruminal acetate-to-propionate is documented [43]. Higher acetate-to-propionate ratios affects both milk fat contents and animal performance. Conversely, higher propionate-to-acetate ratios (i.e., lower acetate-to-propionate ratios) increase growth and nutrient utilization efficiency. In this study, we identified a strong rumen microbial predictor for the acetate-to-propionate ratio, the normalized log abundance ratio of *Sutterella* and *Desulfovibrio*, using the selbal algorithm (R^2^ = 0.868). Moreover, another microbial signature or predictor had an even better predictive power for propionate-to-acetate ratios (R^2^ = 0.919). The latter microbial predictor consisted of three ruminal bacterial taxa: *Desulfovibrio* and an unclassified genus in BS11 (numerator) and *Sutterella* (denominator). As a group of sulfate-reducing, motile, anaerobic bacteria, the genus *Desulfovibrio* is ubiquitous in the rumen and plays an essential ecological role in the rumen microbial community. As a fermenter for hemicellulosic monomers to produce acetate and, the abundance of BS11 can be enriched. At the same time, the diet contains increased levels of woody biomass, including a high percentage of lignin cellulose and hemicellulose [20]. Interestingly, BS11 played an important hub role in the network derived from grass-fed cattle and had a strong and positive association with several unclassified genera in the families Prevotellaceae and Paraprevotellaceae in this study. It is conceivable that either increasing the abundance of *Desulfovibrio and BS11* or decreasing that of *Sutterella*, or both, via proper choice of forage species or by the use of synbiotics in grass-fed cattle production, will result in an increased propionate-to-acetate ratio (or lowered acetate-to-propionate ratio). While further validation is warranted, this rumen microbial predictor has some practical relevance and can be developed as a promising biomarker to guide successful ruminal manipulation for desired performance traits or feed conversion efficiency and reduced methane emission.

## 5. Conclusions

While grass-fed beef is considered healthier than conventional beef, ADG and the age–weight index of cattle raised under the grass-fed regimen are generally significantly reduced, which has important implications in global green-house gas emission and agricultural land use [23,24,25]. In this study, though comparing total SCFA production between the two feeding regimens and investigating the rumen microbial interactions using advanced algorithms, we identified rumen microbial predictors able to distinguish the two feeding regimens strongly correlated with the acetate-to-propionate ratios in the rumen. As measured by the digestible nutrient intake, forage allowance and quality also have a marked effect on the rumen metabolite, microbial features and biomarker effects. Further experimental validation is warranted, and these rumen microbial signatures can be developed as promising biomarkers for efficient rumen manipulation for desired performance traits.

## Figures and Tables

**Figure 1 animals-12-02995-f001:**
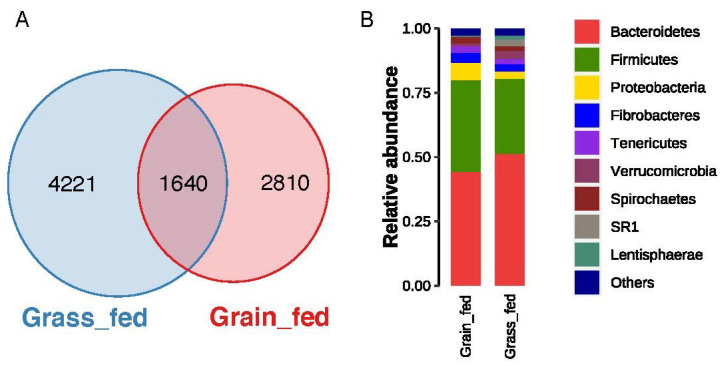
(**A**) Venn diagram of the number of OTUs identified between grass-fed and grain-fed rumen microbiome. (**B**) Stacked bar plot of the relative abundance of rumen bacterial communities at a phylum level between grass-fed and grain-fed group.

**Figure 2 animals-12-02995-f002:**
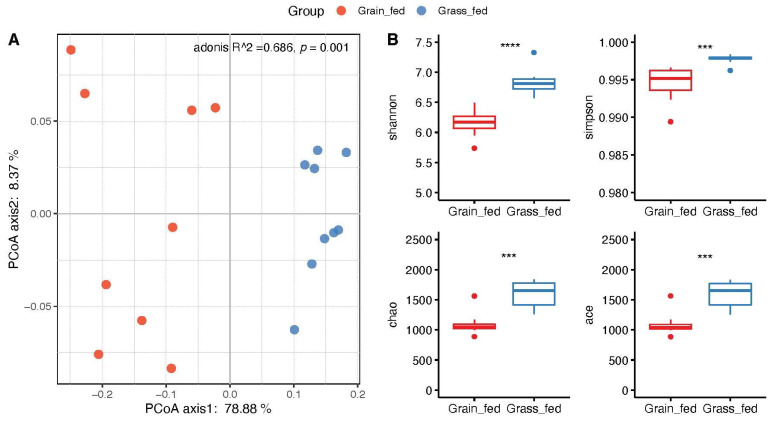
Community diversity analysis of grass-fed and grain-fed rumen microbiome. (**A**) Principal Component Analysis (PCoA) based on Bray–Curtis dissimilarity matrix. (**B**) Alpha diversity indices of bacterial communities between the two groups, including Shannon, Simpson, Chao1 (chao), and Abundance-based Coverage Estimator (ace). **** *p* < 0.001, *** *p* < 0.01.

**Figure 3 animals-12-02995-f003:**
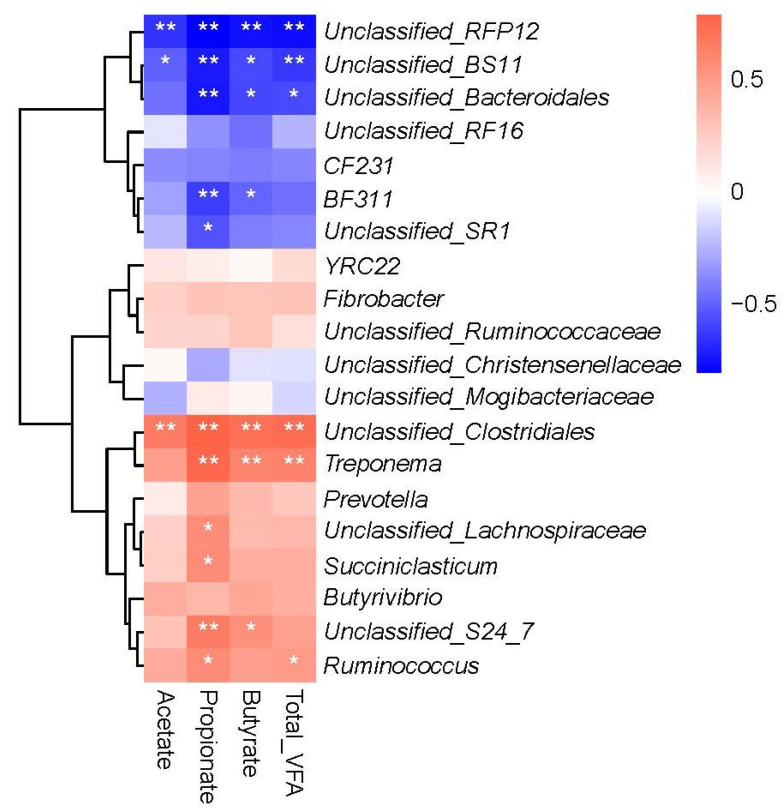
Heatmap showing Spearman correlations between VFAs and top 20 abundant genera. Red color means positive correlation and blue means negative. ** *p* < 0.01, * *p* < 0.05.

**Figure 4 animals-12-02995-f004:**
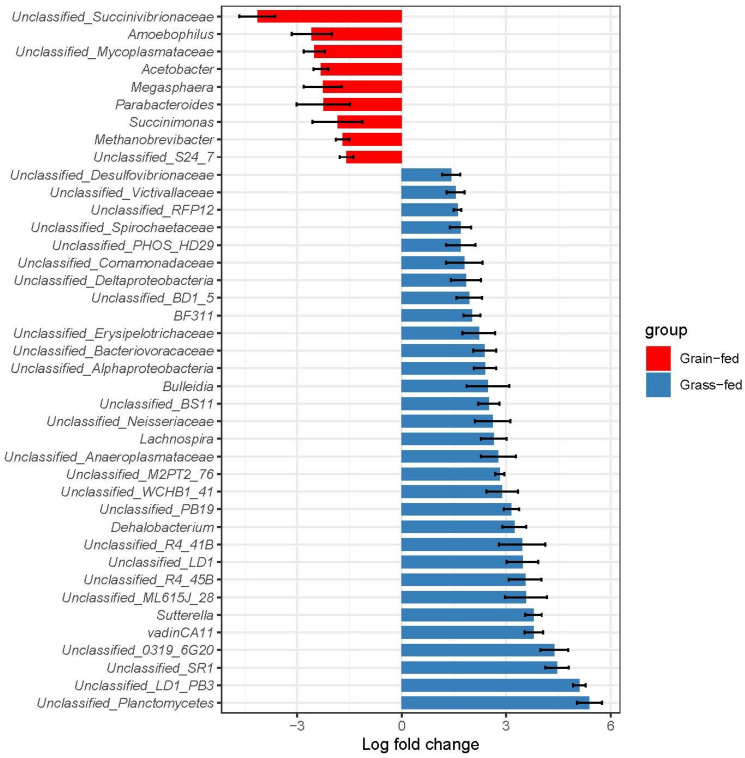
Genera that were significantly different in abundance between the grass-fed and grain-fed rumen microbiome. The *x* axis denotes the log fold change of the abundances for each taxon in grass-fed group over grain-fed group.

**Figure 5 animals-12-02995-f005:**
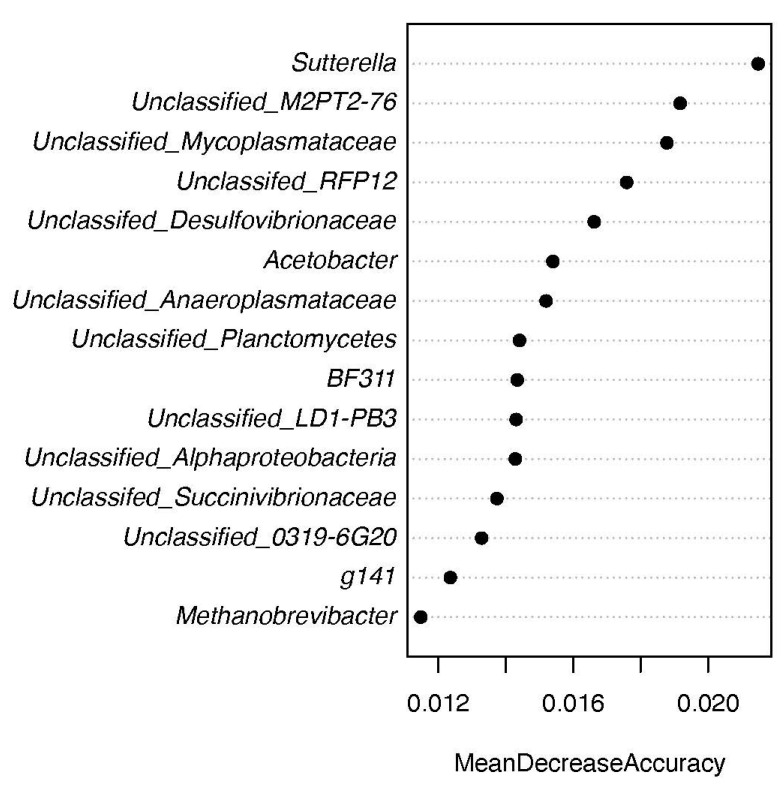
Random Forest analysis on remen bacterial communities of grass-fed and grain-fed Angus cattle. The *y* axis, from top to bottom, displays the top 15 genera ranked by their relative importance based on Mean Decrease Accuracy in the classification of diets.

**Figure 6 animals-12-02995-f006:**
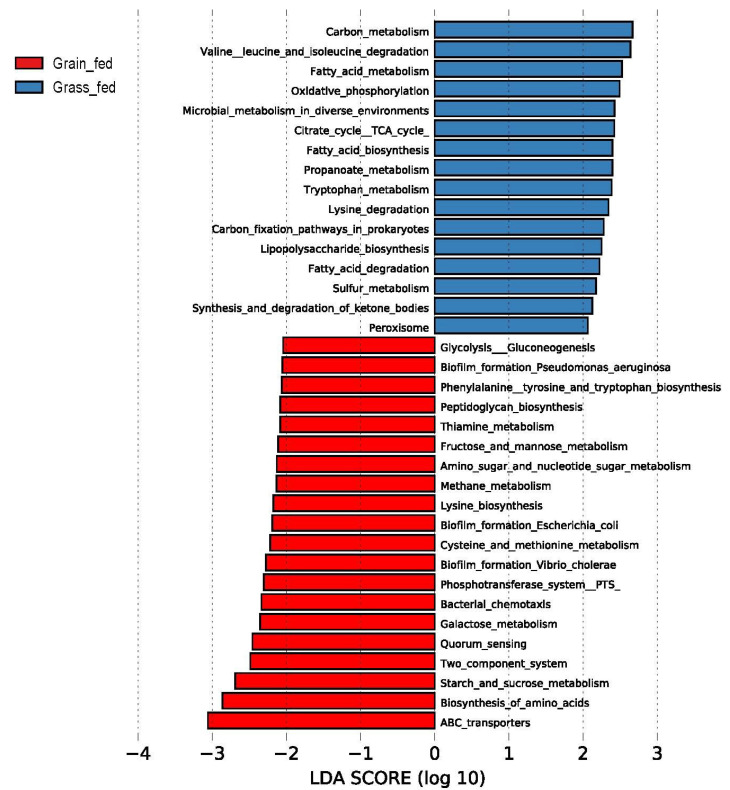
Predicted KEGG pathways significantly influenced by the effect of diets on rumen bacterial communities in Angus beef cattle (LDA score ≥ 2.0).

**Figure 7 animals-12-02995-f007:**
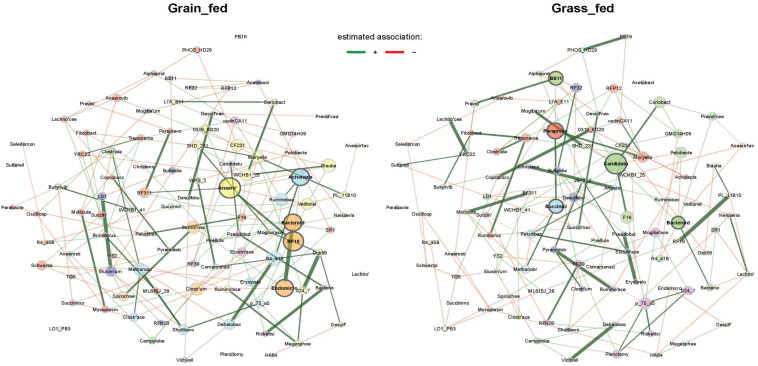
Comparisons of rumen microbial association networks at genus level between grass-fed and grain-fed group. The layout computed for grain-fed network was applied in both networks for comparison. Unconnected nodes in both groups were removed. Genera names were abbreviated.

**Figure 8 animals-12-02995-f008:**
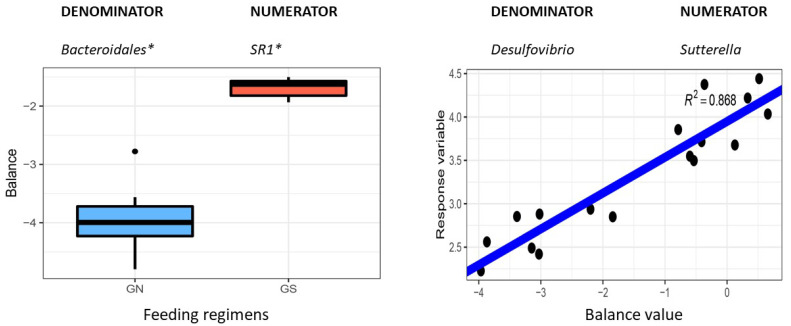
Rumen microbial signatures selected by the selbal algorithm with a strong predictive accuracy. (**A**): The microbial predictor accurately distinguished the grass and grain feeding regimens. (**B**): A rumen microbial signature strongly associated with the ruminal acetate-to-propionate ratio. GS: grass-fed; GN: grain-fed. * unclassified genus assigned to the taxon. Balance values are defined as normalized log abundance ratios of the numerator/denominator.

**Table 1 animals-12-02995-t001:** Analysis of VFA and lactate levels between grass-fed and grain-fed rumen. Concentration unit is shown as parts per million (ppm).

VFA	Grain-Fed	Grass-Fed	*p* Value
Acetate	3990.125 ± 420.756	3197.556 ± 883.497	0.0338 *
Propionate	1515.625 ± 214.071	834.889 ± 296.516	<0.0001 ***
Butyrate	773.125 ± 90.044	539.444 ± 200.4	0.0088 **
Isobutyrate	119.5 ± 10.61	69.111 ± 8.638	<0.0001 ***
Lactic acid	8.286 ± 6.264	2.667 ± 3.162	0.0605 .

Significance codes: ***: 0.001, **: 0.01, *: 0.05, .: 0.1.

**Table 2 animals-12-02995-t002:** Global network metrics and centrality measures of the grain-fed and grass-fed networks. The abs.diff. denotes the absolute difference between the measures of the two groups. *p* Values were adjusted for multiple testing with an adaptive Benjamini–Hochberg method.

	Grain-Fed	Grass-Fed	Abs.Diff.	*p* Value
Global network measures:				
Average path length	2.761	2.863	0.102	0.527473
Clustering coefficient	0.052	0.12	0.067	0.074925 .
Modularity	0.494	0.586	0.092	0.002997 **
Vertex connectivity	1	1	0	1
Edge connectivity	1	1	0	1
Edge density	0.038	0.037	0.001	0.818182
Degree (normalized):				
Lachnospira	0	0.061	0.061	0.620445
Candidatus Amoebophilus	0.03	0.081	0.051	0.620445
Unclassified_LD1-PB3	0	0.051	0.051	0.620445
Unclassified_Planctomycetes	0	0.04	0.04	0.620445
Acetobacter	0.04	0	0.04	0.802088
Desulfobulbus	0.03	0.071	0.04	0.620445
Unclassified_Lachnospiraceae	0.051	0.01	0.04	0.620445
Anaeroplasma	0.071	0.03	0.04	0.620445
Clostridium	0.061	0.02	0.04	0.620445
SHD-231	0.051	0.01	0.04	0.620445
Eigenvector centrality (normalized):				
Anaeroplasma	1	0.09	0.91	0.097113 .
Unclassified_RF16	0.949	0.058	0.891	0.218505
Candidatus Amoebophilus	0.166	1	0.834	0.58268
Unclassified_Acholeplasmatales	0.928	0.098	0.831	0.097113 .
Unclassified_Endomicrobia	0.72	0.017	0.703	0.097113 .
Unclassified_RFP12	0.132	0.701	0.569	0.776907
Clostridium	0.575	0.056	0.519	0.49944
Blautia	0.515	0.026	0.489	0.49944
Treponema	0.171	0.659	0.488	0.794249
Unclassified_Veillonellaceae	0.501	0.014	0.488	0.49944

Significance codes: **: 0.01, .: 0.1.

**Table 3 animals-12-02995-t003:** Jaccard index values of the most central nodes and hub taxa between the two networks of grass-fed and grain-fed rumen microbiome.

	j	*p* (J ≤ j)	*p* (J ≥ j)
degree	0.088	0.000934 ***	0.999837
betweenness centr.	0.19	0.031715 *	0.986817
closeness centr.	0.19	0.031715 *	0.986817
eigenvec. centr.	0.163	0.010359 *	0.996387
hub taxa	0.111	0.143068	0.973988

Significance codes: ***: 0.001, *: 0.05.

## Data Availability

The data generated for this study can be found in the NCBI Sequence Read Archive (SRA) under BioProject PRJNA758549.

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
