# Peer review of "Rumen Microbial Predictors for Short-Chain Fatty Acid Levels and the Grass-Fed Regimen in Angus Cattle"

_animals, 2022, doi:10.3390/ani12212995_

Round 1

Reviewer 1 Report

The work is interesting and the results have the potential as biomarkers for efficient rumen manipulation for desired performance traits. I recommend this paper to be published after the following minor modifications. 

1.      Please provide the age of the experimental cattle, and their weight at the time of slaughter.

2.      The authors are advised to add details about the feeding experiment, such as how long it lasted, what month or season it started.

3.      Line184: “**** p < 0.001, *** p < 0.01.” “p” needs to be italic. Please check and correct in the full text.

4.      Line 198-200: “Intriguingly, several standard and core rumina microbiome components, such as Prevotella, Butyrivibrio, and Fibrobacter, were not correlated with any SCFA measurements.” Is this result consistent with your expectations? What should be the explanation?

Author Response

Q1: Please provide the age of the experimental cattle, and their weight at the time of slaughter. Q2: The authors are advised to add details about the feeding experiment, such as how long it lasted, and what month or season it started.

A1 and A2: The date of birth, birth weight, dam, and sire information in spring were recorded.  Body weight was measured every 24 to 28 days to calculate average daily gain (ADG). After reaching the market weight of around 1000 lbs., the Angus steers, grass-fed animals at 22 months old and grain-fed animals at 16 months old, were slaughtered; and the liquid rumen samples were immediately collected and stored at −80 °C for microbial DNA extraction.

A3:  Line184: “**** p < 0.001, *** p < 0.01.” “p” needs to be italic. Please check and correct in the full text.

A3: We corrected it.

Q4. Intriguingly, several standard and core rumina microbiome components, such as Prevotella, Butyrivibrio, and Fibrobacter, were not correlated with any SCFA measurements.

A4. The result indicates that we need to study further.

Reviewer 2 Report

Forage allowance and quality, as measured by the digestible nutrient intake, has a marked effect in the rumen metabolite, microbial features, and biomarkers effects, so we suggest that the authors consider these factors in this paper.

Considering the information described in the methodology, it is necessary to change the title, as well as all the terminology used in the article, because alfalfa is a legume, so the free-range grass-fed denomination using legume for grazing and supplementation of animals is not correct.

Add the data regarding the chemical composition, digestible energy values of the feed used in the experiment, since the intake of digestible nutrients are determinants of the rumen metabolite, microbial features, and biomarkers.

Authors discuss and conclude about age-weight index data, feed conversion ratios (FCR), but do not present the values obtained in this research. Likewise, in the conclusion, there is a reference to greenhouse gas emissions, data not evaluated in this research. We recommend using the support of specific citations and presenting this discussion in the discussion session.

Maintain this information: In this study, though comparing total SCFA production between the two feeding regimens and investigating the rumen microbial interactions using advanced 489 algorithms, we identified rumen microbial predictors able to distinguish the two feeding regimens strongly correlated with the acetate to propionate ratios in the rumen. While further experimental validation is warranted, these rumen microbial signatures can be developed as promising biomarkers for efficient rumen manipulation for desired performance traits.

Author Response

Q1: Forage allowance and quality, as measured by the digestible nutrient intake, has a marked effect on the rumen metabolite, microbial features, and biomarkers effects, so we suggest that the authors consider these factors in this paper.

A1: As measured by the digestible nutrient intake, forage allowance and quality also have a marked effect on the rumen metabolite, microbial features, and biomarker effects. Further experimental validation is warranted; these rumen microbial signatures can be developed as promising biomarkers for efficient rumen manipulation for desired performance traits.

Q2: Authors discuss and conclude about age-weight index data, and feed conversion ratios (FCR), but do not present the values obtained in this research. Likewise, in the conclusion, there is a reference to greenhouse gas emissions, data not evaluated in this research. We recommend using the support of specific citations and presenting this discussion in the discussion session.

A2: We have cited references as follows:

44. Arndt C, Misselbrook TH, Vega A, Gonzalez-Quintero R, Chavarro-Lobo JA, Mazzetto AM, Chadwick DR: Measured ammonia emissions from tropical and subtropical pastures: A comparison with 2006 IPCC, 2019 Refinement to the 2006 IPCC, and EMEP/EEA (European Monitoring and Evaluation Programme and European Environmental Agency) inventory estimates. J Dairy Sci 2020, 103(7):6706-6715.

  1. Oliveira PPA, Berndt A, Pedroso AF, Alves TC, Pezzopane JRM, Sakamoto LS, Henrique FL, Rodrigues PHM: Greenhouse gas balance and carbon footprint of pasture-based beef cattle production systems in the tropical region (Atlantic Forest biome). Animal 2020, 14(S3):s427-s437.
  2. Andrade BGN, Bressani FA, Cuadrat RRC, Cardoso TF, Malheiros JM, de Oliveira PSN, Petrini J, Mourao GB, Coutinho LL, Reecy JM et al: Stool and Ruminal Microbiome Components Associated With Methane Emission and Feed Efficiency in Nelore Beef Cattle. Front Genet 2022, 13:812828.

Reviewer 3 Report

This study examined the effect of grass- vs. grain-fed on the microbiome and rumen fermentation profile of beef steers. Researchers characterized the microbial interactions in the rumen using advanced algorithms and compared global microbial networks between grass- and grain- fed production systems. I think the study has some significance for beef cattle production, and the manuscript is well written and suitable for publication.

Suggestions:

1. Diet formulations for the experimental treatments need to be given.

2. The authors need to give a clear purpose of the trial. For grass-fed and grain-fed cattle, meat quality is paramount, and the relationship between differences in rumen microbes and meat quality needs to be emphasized.

3. Author could scrred the rumen metabolite, microbial features, and biomarkers with high predictive power for grass-fed cattle, but how should they be applied in beef cattle production in the future?

4. Image clarity needs to be improved.

Author Response

Q1. Diet formulations for the experimental treatments need to be given.

A1: Diet formulations for the experimental treatments were described [11]. 

Q2. The authors need to give a clear purpose of the trial. For grass-fed and grain-fed cattle, meat quality is paramount, and the relationship between differences in rumen microbes and meat quality needs to be emphasized.

A2. Although we found the differences in beef quality and metabolites between the two diet styles, the similarity and difference in rumen microbial composition and species interaction between grass-fed and grain-fed cattle have not been systemically evaluated. Furthermore, the metabolite and microbial predictors or biomarkers associated with the grass-fed regimen have yet to be identified. 

Q3. The author could scrred the rumen metabolite, microbial features, and biomarkers with high predictive power for grass-fed cattle, but how should they be applied in beef cattle production in the future?

A3: These rumen microbial signatures can be developed as promising biomarkers for efficient rumen manipulation for desired performance traits.

Q4. Image clarity needs to be improved.

A4: They are improved.

Reviewer 4 Report

It is interesting manuscript, generally well written. Nevertheless, some details must be added or crarified before acceptation.

Material and methods: Please provide:

 the age of animals at the begining of the experiment

The duration of experiment – how long the animals were fed.

The reason of primer choise for exactly tacson 341/357F. There are other options, why You did not use primers based on paper: doi: 10.3389/fmicb.2017.00494 by Thijs et al. 2017?

Author Response

Q1: the age of animals at the begining of the experiment

A1: After weaning, the animals are grouped.

Q2: The duration of the experiment – how long the animals were fed.

A2: The grass-fed cattle are fed for 22 months, while grass-fed ones fed 16 months.

Q3: The reason of primer choise for exactly tacson 341/357F. There are other options, why You did not use primers based on paper: doi: 10.3389/fmicb.2017.00494 by Thijs et al. 2017?

A3: We will use the primes from Thijs et al in the future.

Round 2

Reviewer 2 Report

The experiment was conducted evaluating treatments related to alfalfa, a forage legume, therefore the term grass-fed regimen is inappropriate.

Line 76 to 78: “We also attempted to identify rumen metabolite, microbial features, and biomarkers with high predictive power for grass-fed cattle. Our findings will have pragmatic implications for enhancing production efficiency in the grass feeding regimen”. Forage allowance and quality, as measured by the digestible nutrient intake, has a marked effect in the rumen metabolite, microbial features, and biomarkers effects, so we suggest that the authors consider these factors in this paper.

Line 81 to 87: “Eighteen steers at the Wye Angus beef cattle herd of the University of Maryland were used for this study. Two types of feeding methods, grain-fed and free-range grass-fed, were used in this herd as two experimental groups. The grain-fed group received a finishing diet consisting of silage corn, shelled corn, soybeans, and trace minerals. The grass-fed group had free access to grazed alfalfa and alfalfa, bailage during the cold season. The alfalfa crop did not use fertilizers, pesticides, or other artificial chemicals. Grass-fed cattle in this herd were not fed any animal, agricultural or industrial by-products and received no grain”. Considering the information described in the methodology, it is necessary to change the title, as well as all the terminology used in the article, because alfalfa is a legume, so the free-range grass-fed denomination using legume for grazing and supplementation of animals is not correct. Add the data regarding the chemical composition, digestible energy values of the feed used in the experiment, since the intake of digestible nutrients are determinants of the rumen metabolite, microbial features, and biomarkers.

Line 85: Define the storage process, dry matter content of alfalfa bailage. Check the term alfalfa bailage.

In conclusion session there is a reference to greenhouse gas emissions, data not evaluated in this research. We recommend using the support of specific citations and presenting this discussion in the discussion session.

The experiment was conducted evaluating treatments related to alfalfa, a forage legume, therefore the term grass-fed regimen is inappropriate.

Author Response

Q1 : Line 76 to 78: “We also attempted to identify rumen metabolite, microbial features, and biomarkers with high predictive power for grass-fed cattle. Our findings will have pragmatic implications for enhancing production efficiency in the grass feeding regimen”. Forage allowance and quality, as measured by the digestible nutrient intake, has a marked effect in the rumen metabolite, microbial features, and biomarkers effects, so we suggest that the authors consider these factors in this paper.

A1: Our findings will provide deep insight into enhancing production efficiency in the grass feeding regimen.

Q2: Line 81 to 87: “Eighteen steers at the Wye Angus beef cattle herd of the University of Maryland were used for this study. Two types of feeding methods, grain-fed and free-range grass-fed, were used in this herd as two experimental groups. The grain-fed group received a finishing diet consisting of silage corn, shelled corn, soybeans, and trace minerals. The grass-fed group had free access to grazed alfalfa and alfalfa, bailage during the cold season. The alfalfa crop did not use fertilizers, pesticides, or other artificial chemicals. Grass-fed cattle in this herd were not fed any animal, agricultural or industrial by-products and received no grain”. Considering the information described in the methodology, it is necessary to change the title, as well as all the terminology used in the article, because alfalfa is a legume, so the free-range grass-fed denomination using legume for grazing and supplementation of animals is not correct. Add the data regarding the chemical composition, digestible energy values of the feed used in the experiment, since the intake of digestible nutrients are determinants of the rumen metabolite, microbial features, and biomarkers.

Line 85: Define the storage process, dry matter content of alfalfa bailage. Check the term alfalfa bailage.

A2: The chemical composition, and digestible energy values of the feed used in the experiment was described [11]. 

Q3: In conclusion session there is a reference to greenhouse gas emissions, data not evaluated in this research. We recommend using the support of specific citations and presenting this discussion in the discussion session.

A3: When this consumption trend is expected to continue, increased environmental burden [27, 28], including higher methane emissions [29], resulting from such a swift should be carefully considered.